# Determining a Welfare Prioritization for Horses Using a Delphi Method

**DOI:** 10.3390/ani10040647

**Published:** 2020-04-09

**Authors:** Fiona C. Rioja-Lang, Melanie Connor, Heather Bacon, Cathy M. Dwyer

**Affiliations:** 1Royal (Dick) School of Veterinary Studies, University of Edinburgh, Edinburgh EH25 9RG, UK; Fiona.lang@ed.ac.uk (F.C.R.-L.); melanie.connor@ed.ac.uk (M.C.); Heather.Bacon@ed.ac.uk (H.B.); 2Animal Behaviour and Welfare, Department of Animal and Veterainry Sciences, Central Faculty, Scotland’s Rural College (SRUC), Edinburgh EH9 3JG, UK

**Keywords:** horse, welfare, Delphi method, health, behaviour, management, training, nutrition

## Abstract

**Simple Summary:**

Horses are used for a wide range of different purposes and may be vulnerable to a large number of different welfare issues, some of which are unique to equines, such as responses to being ridden or trained. Compared to farmed livestock, their welfare has received less attention, but concern for their welfare is increasing. Welfare issues can arise from the environments in which animals are kept, how they are treated by their human caregivers and their health. To determine which of the issues are most important and may need the greatest attention in terms of research effort or owner education, we used a process of eliciting expert opinion. Through a series of surveys and ranking of issues, we determined that, in the opinion of equine experts, the most important issues for horses were poor disease prevention, issues arising when old or sick horses are not promptly euthanized, lack of owner knowledge of welfare needs of horses, fear and stress involved in horse use, inability of owners to recognize pain behaviour, obesity and inadequate feeding practices. Prioritizing different welfare issues can help to focus attention on the most pressing or severe issues causing the greatest amount of suffering.

**Abstract:**

Equine welfare issues are receiving increasing attention in the UK, but welfare problems can arise from a wide range of causes. In order to identify the most important welfare concerns for horses, we used a Delphi method with 19 equine welfare experts. An initial list of 84 equine welfare issues was generated using an online discussion board and NVivo thematic analysis. Subsequently, experts ranked these welfare issues for perceived prevalence, severity and duration of suffering associated with each issue on a 6-point Likert scale. All issues with a mean score of 3 or above (*n* = 37) were included in subsequent rounds. Finally, a subset of experts attended a two-day workshop to determine the final priority list of welfare issues. The welfare issues perceived to be most prevalent were lack of biosecurity, delayed euthanasia, lack of owner knowledge of equine welfare needs, fear and stress from use, and obesity. The issues considered to cause greatest suffering for individual horses were delayed euthanasia, lack of recognition by owners of pain behaviour, large worm burdens, obesity and being fed unsuitable diets for equine feeding behaviour. These outcomes can help to focus research and education interventions on the most pressing welfare issues for horses.

## 1. Introduction

Survey studies and estimates suggest that there are around 7 million horses in the EU [1], with approximately one million horses in the UK (850,000 horses in England and Wales [2], 100,000 in Scotland [3] and approximately 160,000 in Ireland [4]. Although the majority of horses are kept as leisure animals, horses are unusual in that they are often not classified as companion animals (for example, only 58% of horse owners classified horses as companion animals in a survey in Illinois [5]) and many do not live at the same address as their owners, but neither are they considered as livestock. In addition, although few horses in Europe are now used for draught purposes, a number of horses are also kept for racing, competitions and other professional uses [4] and are used in tourism, forestry, agriculture and therapy [1]. Globally, horses and other equids are still widely used for traction power in low- and middle-income countries [6], are used for meat in many countries and for conservation grazing, and there are also feral horse populations. Thus, there are multiple roles that are occupied by equids, which can carry different risks for horse welfare. In the UK, regardless of whether for professional or amateur use, most horses are ridden and trained by their owners, which can also result in particular welfare issues not experienced by other animals (e.g., bit lesions, aggressive riding style [7,8]).

Until relatively recently, horse welfare studies had focused primarily on stereotypic behaviour, horse transport and health-related issues. However, there has been a recent increase in the number of papers addressing varied horse welfare issues, considering the management of horses (such as pasture access [9], feeding strategies [10], rugging [11] and unwanted horses [5]), horse behaviour [12], horse welfare assessment [13] and particularly issues with the ridden horse and training methods [14,15]. These studies suggest that there are a significant number of horse welfare issues that may be overlooked by their owners, or are related to cultural or traditional methods of horse ownership and training. 

Where there are many potential welfare issues (as have been suggested in previous studies [1,16,17,18,19]), it can sometimes be necessary to identify and prioritize the more important welfare issues for further research and/or education programs. A number of previous studies have used different types of qualitative techniques to determine what those who work with horses considered the most important welfare issues. Butler et al. [16] used focus groups of participants with experience of the racehorse industry to determine welfare priorities for racehorses and concluded that health issues were considered the most important. DuBois et al. [17] used a Delphi technique to identify the perceived prevalence of welfare issues for horses in Canada, suggesting that horses being denied access to important psychological or physical resources were the most important issues. Using qualitative interviewing approaches of various groups involved with horses, Horseman et al. [18] highlighted 45 welfare issues that potentially affected horses in GB, which were focused on health, management and training issues. A Delphi technique, with vignettes, was also used by Collins et al. [19], which particularly identified disposal of horses and behaviour at unregulated gatherings as issues for horses in Ireland.

Use of expert opinions, using methods such as Delphi, focus groups or semi-structured interviews, can be an efficient way to determine welfare priorities, particularly when there is limited empirical scientific data on which to base decisions. However, one of the limitations of this approach can be the repeatability of these studies, and whether another group of experts would reach the same conclusions and priorities. In addition, prioritization may be made for a number of different reasons, such as the feasibility of achieving an improvement in welfare. The aim of this study was to use a Delphi method firstly to gain a comprehensive understanding of the range of welfare issues experienced by horses in the UK (using a Delphi conference), and then to determine a prioritization following from the premise that, where the animals’ capacities to suffer can be assumed, the most pressing issues are determined by the severity and duration of suffering and the number of animals affected [20]. Finally, the prioritization of welfare issues derived from our study is compared to other studies to address whether expert opinion is a suitable method for determining welfare priorities for horses. The results reported here are part of a larger study to determine the priority welfare issues for a range of different managed species in the UK. 

## 2. Materials and Methods 

All research generated from this study was approved by the University of Edinburgh’s Human Ethics Review Committee (HERC). The study formed part of a larger study to identify and prioritise welfare issues for a range of different species. Detailed methods were as previously described [21] and are given in brief here.

### 2.1. Recruitment of Experts

The aim was to recruit between 12 and 20 horse welfare experts to the study and to recruit a broad range of stakeholders, including practising veterinarians, academics, trainers, charity sector employees and equine industry representatives. Experts were defined as having worked in their field for at least 3 years and being based in the UK. Experts were recruited using a snowball sampling technique and were sent a consent form to sign in accordance with HERC guidelines before commencing the study. 

### 2.2. Horse Welfare Issues

A list of horse welfare issues was developed through the use of a Delphi Conference procedure using an online discussion board. An initial list of horse welfare issues (derived from a list generated by the British Veterinary Association) was provided and experts were then able to add to and comment on issues anonymously for two weeks. All comments and discussion from the online platform were collated, and a detailed thematic analysis was conducted using NVivo 11 Pro. Using an emergent coding process, each comment from the board was categorized into themes, reaching saturation at 12 themes, which were later combined into 9 categories. Duplicate welfare issues were deleted, and the final list was checked by two independent assessors to remove redundancies or to add omissions. 

### 2.3. Questionnaires

Two rounds of anonymous surveys were conducted using the Online Survey tool (formerly Bristol Online Survey, JISC, Bristol, UK). In both rounds, demographic data were collected from the experts including age, gender, expertise (experts could indicate expertise in more than one category; for example, they may be both veterinarians and academic researchers) and highest level of educational attainment. 

For the first survey, participants were asked to score each of the potential welfare issues derived from the thematic analysis for severity (the intensity of suffering likely to be a consequence of the welfare issue in their opinion), duration (the estimated time period over which an animal was likely to experience the welfare issue in their opinion) and perceived prevalence (the likely proportion of the UK horse population they considered to be affected by the welfare issue). Each issue was scored on a 6-point Likert scale for each factor, where 1 = never/none, and 6 = always/high. An even-numbered scale was chosen as this forced the experts to make a choice (prioritised or not). 

The results of the first survey were reviewed and mean scores for each welfare issue and each factor calculated. Those issues that scored at least 3.0 or above (i.e., were considered by the experts to be at least somewhat important for any of the three welfare factors) were included in the second survey as a ranked list. In the second round, experts were asked whether they agreed or not with the ranking position, and whether, in their opinion, the issue should be ranked higher or lower. Agreement between experts was assessed by calculating Fleiss’ kappa statistics and by assessing the proportion of experts that agreed with each rank position. 

### 2.4. Workshop

The final stage of the process was an expert workshop conducted in Edinburgh over 2 days. This involved two horse welfare experts drawn from the experts who completed the earlier rounds, as well as 19 other experts from the wider study who had broad animal welfare expertise in other species, including farm and companion animals (overall workshop participants consisted of academics = 6, veterinarians = 5, industry representatives = 4, charity/NGOs = 6). Over the two days, experts participated in small-group (species-specific) and larger-group exercises to achieve a final rank order, working from the ranked prioritized lists of horse welfare issues that had been generated in the earlier rounds. In a few places, experts also combined areas which they felt to be too similar to separate. Consensus at the workshop was considered to be unanimous where a final list was produced with the agreement of all participants.

## 3. Results

### 3.1. Expert Demographics

Nineteen experts were recruited to the horse study. Experts had a mean age of 45 (SD = 7.89) and were predominantly female (17 female, 2 male). Experts were associated with NGOs or equine charities (28%) or were academic researchers (26%), trainers (18%), industry representatives (13%), veterinarians (8%), associated with equine policy (5%) or had other expertise (2%). Experts were nearly all educated to at least university graduate level (94%), with 53% holding postgraduate qualifications (18% Masters, 35% PhD). The response rate was 68% (first round) and 74% (second round). 

### 3.2. Horse Welfare Issues

Horse welfare experts generated a comprehensive list of 84 welfare concerns from the analysis of the discussion board. These were finally classified into 9 themes in preparation for the first online survey (Table 1).

### 3.3. Expert Rankings

The outcome of the first ranking process conducted by experts is shown in Table 2. Of the initial 84 welfare issues shown in Table 1, 34, 37 and 35 issues were retained into round 2 for perceived prevalence, severity and duration of the welfare issue respectively.

Horse experts had an overall low level of agreement with the rank order generated in the first round survey (Fleiss’ kappa = 0.259, 0.227 and 0.243 for perceived prevalence, severity and duration respectively), and generally less than 75% of experts agreed on the top 10 highest-ranking welfare issue for perceived prevalence, severity or duration, although there was better agreement (70%–85%) on the five lowest ranking positions for all welfare issues. Experts considered that lack of understanding of horse welfare needs by owners, over-rugging, inappropriate training and handling, and delayed euthanasia were more prevalent than as scored in the first round. In addition, hunger and obesity were considered less severe, but lack of recognition of pain behaviour, delayed euthanasia and inability to perform normal social behaviours were more severe (see Appendix A for details).

### 3.4. Workshop Rankings

The workshop was more successful than the anonymous online surveys in achieving a consensus (unanimity was achieved), and over two days of discussion, a final welfare ranking on which the experts agreed was achieved (Table 3).

## 4. Discussion

Although the horse experts in the initial rounds of the Delphi did not achieve very high levels of agreement, a better consensus was achieved during the face-to-face discussions in the workshop, and many of the areas considered to be important in the survey were also ranked highly in the workshop. Overall concerns reflected all aspects of horse management and use, from health (biosecurity, worm burdens and delayed euthanasia decisions), owner knowledge (lack of recognition of pain behavior and lack of understanding of equine welfare needs), management (diets, weaning and social groups) and use of horses (fear and frustration, overwork, poorly fitting tack and inappropriate rider weight). The initial rounds of the Delphi were anonymous and had the advantage that individual respondents were not influenced by the decisions of others. However, a disadvantage of the Delhpi method is that it prevents live discussion where individual ideas and perceptions can be broken down, discussed and reassessed. In our study, we found that considered discussion, in a structured way in the workshop, allowed a better consensus to be achieved, building on the preliminary analyses derived in the anonymous process. 

Other qualitative studies that have considered horse welfare have used varying techniques, and not all studies yielded outcomes that could be directly compared to our prioritization. However, a UK study using interviewing and focus groups of industry professionals [18] found that horses being underweight or overweight, poor foot care, internal worms and laminitis were the most frequently mentioned horse health issues. Large worm burdens, obesity and hunger all also appeared in the most important issues in our study for individual horses. Likewise, prolonged stabling, under- or over-feeding, inappropriate rugging and social isolation were identified in previous work [18] and were also considered important in our study for individual horses, and feeding of unsuitable diets was perceived to be prevalent. Use of poorly fitting tack was identified in both studies as a significant cause for welfare concern. In the study of Horseman et al. [18], the most prevalent welfare issues (identified as the perception by stakeholders to be the situations in which welfare was most likely to be compromised) were keeping horses in unsuitable environments, inappropriate use, misunderstanding behaviour, disruption of social routines (e.g., through moving yards), abandonment, transport and delayed euthanasia. Of these, issues with delayed euthanasia, lack of owner understanding, issues with use and instability of social routines were also considered by the experts in our study to be prevalent with horses in the UK. A Delphi study conducted in Canada identified lack of owner knowledge, delayed euthanasia decisions, inappropriate training or use, inappropriate feeding, lack of turnout and lack of social companions as important welfare issues at the individual horse level and issues with breeding, delayed euthanasia, biosecurity and owner ignorance as important welfare issues at the population level [17]. Finally, a study using the Five Domains Framework to assess horse welfare issues within different categories [22] concluded that the most severe impacts on horse welfare were abrupt individual weaning, feeding 100% low-energy concentrate, use of indoor tie-stalls without social contact, forced flexion (sometimes called ‘Rollkur’ or hyperflexion [15]), restrictive nosebands, ear twitches and transport, either individually or in a group with unfamiliar companions. A number of veterinary procedures or surgical interventions were also identified as causing severe welfare impacts [22]. However, this study did not assess the likelihood or prevalence of these issues, only the most severe within a set of specific categories. 

Despite use of different qualitative methods, different selection of experts or industry professionals, and different ways of prioritization, there are a number of similar issues that have arisen in UK and Canada that may reflect general areas of concern for the welfare of horses. In our study, we specifically asked experts to address welfare by focusing on the amount of suffering caused by the welfare issue. The study of McGreevy et al. [22] used the Five Domains model, which asked experts to focus on the impact on the mental state of the horses in determining the welfare impact. Although other studies did not appear to use a specific definition of welfare to direct respondents, participants may have utilized similar thinking about the impact on equine mental state and suffering of the welfare issues that were mentioned frequently. Other studies have also identified racehorses or competition horses [17,18] and horses owned by travellers [18] as areas of particular welfare concern. In our study, although these areas were mentioned in the initial rounds of developing the lists of horse welfare concerns, they were not included in the final prioritization. Recent studies of racehorse welfare [16] and perceptions of welfare in travellers [23] have addressed these issues and suggest that some concerns may be misplaced, which perhaps reflects the lower concern for these areas in our study.

### 4.1. Owner Knowledge and Understanding

Of the issues that arose in all studies, owners’ lack of knowledge, particularly in relation to horse behaviour, was clearly highlighted. In our study, this was identified as a specific theme in the initial development of the list of welfare concerns for horses and was also included as one of the most important issues in the final rankings for recognition of pain behaviour and lack of knowledge of horse welfare needs. Lack of owner knowledge may also contribute to other welfare issues, such as inappropriate feeding and training methods and use of restrictive or poorly-fitting tack. In their review of recreational horse management, Hemsworth et al. [24] suggest that a number of horse owner attributes might contribute to poor horse welfare, including commitment to horse ownership, income, education, knowledge of horse husbandry and attitude to horse management. Of these, levels of education and income, as well as lack of knowledge, have been shown to be associated with horses having welfare problems [24]. Hartmann et al. [25] found high agreement amongst Scandinavian horse owners that horses should be kept in groups but also observed that many horses were not kept with social companions, suggesting that factors other than just lack of knowledge, such as opportunity or financial ability to manage horses differently, may also contribute.

A number of studies have suggested that the vast majority of owners believe horses to be sentient and capable of specific emotions such as pain, fear or joy [26,27]. However, DuBois et al. [27] suggest that belief in sentience did not appear to reflect understanding of welfare issues. In a recent survey [28], horse owners could recognize at least some behaviours associated with distress but missed or incorrectly classified behavioural indicators of negative affective states as positive. Further, a number of participants in the survey indicated that they would be happy for their horse to be treated by methods they had already identified as causing distress [28], suggesting that knowledge was not the only factor involved in poor horse welfare. 

The impact of owner knowledge or management responses has also been shown to be an issue with health practices as well as behaviour. Although lack of knowledge or understanding was not specifically assessed, Thompson et al. [29] found that 19% of Australian horse owners did not vaccinate their horses against tetanus, and 11% did not ensure regular dental care. In addition, McGowan et al. [30] demonstrated that owners could not identify all clinical signs of ill health in their aged equines and did not seek veterinary advice in all cases when it was warranted. These data support the view that lack of sufficient owner knowledge can contribute to poor health and welfare in horses. However, other barriers to improving horse welfare may be related to financial constraints, opportunity to change practices, habit and cultural norms around horse management. 

### 4.2. Delayed Euthanasia Decisions

The consequences of delaying euthanasia decisions, and so increasing the risk of animals suffering uncontrollable pain or disease, was considered by experts in this study to be one of the most prevalent horse welfare issues and to cause the greatest suffering to individual horses. Delayed euthanasia was also identified as important in two other studies that have attempted to prioritize horse welfare issues [17,18], suggesting that this is widely considered an important issue for horses. This may be attributable to owners’ lack of knowledge or ability to correctly identify and understand the consequences of clinical symptoms as discussed above, and may also be related to issues with income or ability to pay for horses to be euthanized. Horse slaughter at abattoirs is permitted in the UK, which may alleviate some of the financial impact of euthanasia, and has been suggested to improve horse welfare [31]. However, other issues such as emotional attachment, peer pressure and negative attitudes to death can also play a role [18]. Currently, there is little published research to allow quantification of the impact of delayed euthanasia on horse welfare, although many veterinary clinics and charities do provide guidance on recognizing a deteriorating quality of life (e.g., www.bluecross.org.uk and www.bhs.org.uk) or emergency conditions where euthanasia would be recommended. 

### 4.3. Impact of Inappropriate Training or Use of Horses

Horses are unusual amongst other domesticated species in the degree of use and training that they usually receive, often by owners who themselves have had no specific education in learning theory or training practices. This lack of owner knowledge of equine learning or training methods was a significant concern in our study and in others [17,18,22]. There has been a recent increase in scientific studies assessing the impact of training, tack use and competition on equine welfare. For example, studies have shown significant lesions in the mouths of Finnish trotters and Icelandic horses when used in competition from the action of the bit [7,32]. The use of coercive hyperflexion of the head and neck or Rollkur, for example in dressage horses, has been shown to cause adverse behavioural [15] and physiological indicators [33], suggesting increased stress in these animals compared to working in a less flexed position. Application of the Five Domains model also identified working in Rollkur as one of the most adverse welfare impacts in foundation training [22]. As already described, a recent study [28] has shown that conflict, stress, fear and frustration were identified as present in a variety of different training and riding disciplines, including natural horsemanship and bridleless riding, and that not all horse owners perceived this to be inappropriate for horse handling. In our study, the misclassification of pain-related behavioural responses as ‘naughty’ was considered to be a significant source of equine suffering and suggests that owner knowledge about equine behaviour, attitudes and understanding of animal pain and beliefs about particular rising practices may need to be tackled to improve equine welfare. 

### 4.4. Inappropriate Feeding

Concerns about feeding practices in our study were either about the quantity of feed given (i.e., that horses were fed too much and were overweight, or not enough and thus were thin or hungry) or that the method of feeding management, such as restricted forage access, might cause adverse behavioural consequences for horses. 

Studies in the UK, Sweden, Italy, Germany and the USA suggest that between 25% and 45% of horses and ponies are overweight or obese [13,34,35,36,37]. These studies confirm that this is indeed a prevalent condition as identified by experts in our study. Obesity is associated with insulin resistance, equine metabolic syndrome [38] and increased likelihood of laminitis (a painful condition of the feet that can result in euthanasia), as well as issues with heat stress, reproductive and inflammatory responses [39]. Thus, obesity is a significant risk factor for poor welfare from the painful and debilitating conditions that are predisposed. By contrast, for horses that were not fed enough food in our study, it was the negative feelings associated with lack of food, i.e., hunger, that were identified as the welfare issue, rather than specifically the consequences of being underweight. In survey studies, only 2% of horses were considered in poor body condition in Scotland [34] and 4.5% in older horses (>15 years) in England and Wales [40]. Thus, it seems that hunger from chronic lack of food does not occur at high prevalence in the UK, although the severity of the condition was the predominant issue for experts in this study. Hunger can occur because horses might be left without food for prolonged periods, even if the intention is to reduce weight in obese animals, such as managing overweight horses on a drylot [36]. Bedding investigation and bedding eating have been suggested to be indicators of feeding frustration which may be associated with hunger [41], and increased motivation to access feed is usually considered an indicator of hunger in other species [42]. Periods of food deprivation have also been linked to an increase in gastric ulceration [43] and are a risk factor for the development of stereotypic behaviour [44]. 

Actions to manage the weight of horses can induce hunger and/or frustration in horses, as described above. However, these emotional states may also occur when horses are fed in ways that are very different from their evolved feeding responses. In our study, feeding animals unsuitable diets for equine feeding behaviour was considered by experts to be both an important cause of horses suffering and to be prevalent. Feral horses spend 12–18 hours a day grazing [45], and their digestive system is adapted to a continuous intake of relatively poor quality forage. Feeding management of stabled horses may often utilize feed restriction (as described above), and feeding in ‘meals’ rather than continuous forage. A survey of UK horse owners found that 4% were continuously housed, and 92% had pasture access for an average of between 70 and 126 hours per week [46], suggesting that horses were spending between 25% and 58% of their time stabled. The majority (82%) of horses received additional forage feeds (such as hay) and concentrate feeds (86%), although the frequency of feeds was not reported [46]. The study of McGreevy et al. [22] concluded that feeding a 100% concentrate diet, even if this met the horses’ metabolic needs, had the most adverse impact on welfare through its impact on behaviour and that pasture choice and feeding forage were the least likely to impact welfare. Rochais et al. [10] showed that feeding systems that encouraged a longer period of feeding were also associated with reduced stereotypic responses and more positive behavioural change in horses. 

### 4.5. Inappropriate Environments and Social Behaviour

In our study, issues with social behaviours were considered both to be prevalent (instability of social groups) and to cause significant suffering to individual horses (inability to perform normal social interactions). Prolonged stabling and management in inappropriate environments were identified as welfare issues in the study of Horseman et al. [18]. In our study, it was the impact of housing/management on social contact that was considered the greatest issue, although these issues are often experienced together. The study of McGreevy et al. [22] also concluded that worse welfare was experienced by animals kept either indoors or restricted outdoors (tethered) without social contact and that best welfare was achieved by living outdoors with full social contact. Horses are social animals, and keeping them in groups is widely considered to be important for good welfare [47]. For example, horses have been shown to have a high motivation to maintain full, or partial social contact with another horse [48], and stereotypic behaviour is linked to single housing [49]. 

Various studies suggest that 83%–92% of horses are continuously managed at pasture in groups or have pasture access [25,46,50] but 4%–10% of horses may be continuously housed with little or no exercise opportunities [13,25,46]. This figure was higher for entire male horses, where nearly 40% were found to be kept alone in a Nordic study [25]. Of the stabled horses in a study of Italian and German horse welfare [13], 22% of horses had no physical or visual contact with other horses. These data suggest that a significant percentage of horses are kept in environments that may prevent social contact for at least some of the day and that a small percentage of horses may have no access to other horses for prolonged periods of time. Nearly 20% of stabled horses have been shown to display stereotypic behaviours [13], suggesting that welfare is compromised in these animals. 

## 5. Conclusions

By comparing the outcomes from our data with those of other studies that have attempted similar types of assessments, we have found that there are common welfare concerns that are raised frequently with different groups, which lends greater weight to arguments that these are important issues for horse welfare. In particular, lack of owner knowledge, or application of knowledge to the management of horses, is an important welfare concern, which has also been seen in other species [21,51]. In addition, many of the other issues that have been highlighted may also stem from poor owner knowledge or the application of traditional or culturally mediated methods of managing or using horses. These include poor biosecurity practices, not recognizing pain behaviour, use of poorly fitting or restrictive tack, inappropriate training practices and keeping horses in environments that do not meet their physical, nutritional or behavioural welfare needs. Increasing numbers of studies have demonstrated that these practices can result in fear, frustration or distress in horses, and an effective means to transfer this knowledge to owners and an assessment of the barriers to implementing changes are required. 

One of the limitations of a Delphi procedure, and for other techniques that rely on expert opinion, is that the outcomes reflect only the knowledge and understanding of the experts [52]. However, it is a useful technique when empirical data are not available, or when comparisons of, for example, chronic versus acute conditions are attempted. Although it uses a consensus approach and seeking experts from a range of different backgrounds [53] can improve the issue, it is still a possibility that this does not reflect reality and a different group of experts might achieve a different outcome. However, by comparing our outcomes with other studies that have used similar approaches, we have been able to overcome this issue and hence conclude that a consensus on priority horse welfare issues has been achieved.

## Figures and Tables

**Table 1 animals-10-00647-t001:** Welfare issues for horses (unranked) derived from thematic analysis of the anonymous online discussion boards, sorted for themes.

Category of Concern	Specific Welfare Issue
Health issues	Lack of routine health care (e.g., vaccinations, dental, parasite prevention) Large worm burdens Lack of understanding re: worming/blanket worming, leading to resistance Lack of easy access to medications Lack of easy access to health care Delay in veterinary/professional engagement Lack of biosecurity and disease surveillance Lack of health checks at some ports/entry points, potential introduction of diseases
Owner knowledge or behaviour	Neglect or sub-optimal care Lack of equine knowledge by owner (‘laziness’ to learn or refusal to change behaviour) Cultural influences e.g., ‘letting nature take its course’ Financial restrictions of owner for better livery arrangements/professional assistance Delayed euthanasia e.g., quality-of-life evaluation methods often not implemented Lack of a credible quality of life assessment mechanism Inappropriate re-homing, especially elderly; euthanasia would be more appropriate Lack of confidence in abattoir as an option for end of life Fundamental lack of owner understanding horse’s ethological needs Anthropomorphism Rugging outside horses 24 h/d for half the year—no control if too hot/cold/itchy ‘Well-meaning but ill-informed owners’ (management traditions) Inaccurate portrayal of stallions in our culture (feisty, difficult to handle) Lack of recognition of pain behaviour before it becomes overt (e.g., can be misinterpreted as ‘bolshy’/naughty) Culture of using horses in a utilitarian way Growing population of owners not knowing where to obtain information Owners receiving poor ‘trusted information’ from yard and online Horses used as a status symbol e.g., upper levels of sport may depend on poor welfare approaches Use of horses as status symbols within the traveller community
Nutrition and management practices	Obesity, e.g., due to lack of work, unrestricted grazing Inappropriate diet, in particular, low-fibre, high energy concentrates, restricted access to high-fibre forages Rise in the blanket use of supplements Undernourished (hunger) Lack of access to fresh, clean water (thirst) Feeding methods of stabled horsed contrary to their evolution (for low-energy forage) Turned out 24/7 on green grass (e.g., relative high energy, risk of obesity/laminitis) Poor field management, including ragwort, grazing too rich/poor, mud Unsuitable living environment e.g., unsafe fencing
Work and training/horse use	Overworking young horses Overweight riders or inappropriate for the horse/pony Horses being competed which are unfit with unfit riders Hirelings e.g., Scottish Common riding equines overused by novice/unbalanced riders Injury during use, e.g., from racing, eventing, endurance events Road racing horses (arguably not a traditional ‘traveller/gypsy’ activity) Being ridden—not poor riding or training, but riding per se Injury to hirelings (overweight riders, ill-fitting tack, overwork) Unfit and unsuitable horses hired out Inappropriate training methods e.g., punishment, negative reinforcement (bit or leg pressure not removed), unclear signals, pain, flooding Public unable to interpret/replicate some training methods e.g., natural horsemanship Inappropriate and restrictive tack, e.g., poorly fitted saddle, certain bits/nosebands
Horse behaviour	Fear and stress arising from use (work, sport, entertainment), e.g., shows, racing, polo Poor handling and training: confusion, conflict, frustration, distress, stress, fear and sometimes pain as a result of the interactions they have with people Lack of understanding of learning theory leading to poor training techniques Poor weaning methods Stabling 24/7—lack of natural behavioural expression due to restrictions Social isolation (no or limited contact with own species) Being kept in a herd that is constantly changing (unstable social groups) Being kept where an individual animal cannot escape from aggressive animals Being kept in group where dominant animals restrict access to resources Restrictions on normal behaviours to satisfy basic dietary needs Lack of environmental control/frustration—horse is motivated to act but cannot Management of stallions—often isolated, unable to perform normal social interactions Negative affective states e.g., atypical myopathy, fear, frustration, depression, anxiety Boredom, even with access to turnout—usually a square monoculture field
Service providers	Unqualified service providers, e.g., unqualified farrier, dentistry, castration not by vet Poor hoof care e.g., lack of care or trimming by owner (inappropriately) Lack of regulation of trainers Lack of animal establishment licensing No regulation of small rescue establishments (sometimes hoarding) Tendency (of industry) to tackle symptoms, not cause (e.g., livery layout, routines) The passport process: not fit for purpose Lack of licensing, inspection, or agreed standards of care in equine establishments (e.g., livery yards, dealers, sanctuaries and rescues)
Transport of horses	Long-distance transport Some domestic/local journeys are made in inappropriate vehicles (unsafe) Little regard for fitness (of horse) to travel
Stray or unwanted horses	Stray and abandoned horses (injuries to themselves and others) Lack of enforced, mandatory Microchips—leads to abandonments Fly grazing Tethering without provision of adequate shelter, attention, water, etc
Breeding and horse trading	Breeding decisions e.g., exaggerated conformation Breeding low-value horses with conformational/physiological compromising conditions Broodmares being bred indiscriminately—i.e., ‘given a job to do’ when retired Indiscriminate breeding and failing to castrate colts Selling horses online—ill-prepared potential owners buying unsuitable horses

**Table 2 animals-10-00647-t002:** Mean scores (with SD) and rank order for horse welfare issues that scored at least mean 3.0 or above for perceived prevalence, severity and duration of the welfare issue (issues scored using a Likert scale from 1–6, where 1 = never/none, and 6 = always/high).

Rank Order	Perceived Prevalence	Mean Score (SD)	Severity	Mean Score (SD)	Duration	Mean Score (SD)
1	Lack of recognition of pain behaviour (e.g., can be misinterpreted as naughty)	4.88 (1.08)	Hunger	5.25 (0.87)	Obesity (e.g., due to lack of work, unrestricted grazing)	5.00 (1.00)
2	Use of inappropriate training and handling	4.82 (1.08)	Lack of access to clean/fresh water	5.08 (1.62)	Lack of recognition of pain behaviour (e.g., can be misinterpreted as naughty)	4.96 (1.04)
3	Obesity (e.g., due to lack of work, unrestricted grazing)	4.75 (0.75)	Obesity (e.g., due to lack of work, unrestricted grazing)	4.92 (1.38)	Hunger	4.60 (1.35)
4	Delayed euthanasia decisions	4.58 (0.90)	Lack of recognition of pain behaviour (e.g., can be misinterpreted as naughty)	4.79 (1.13)	Inability to perform normal social interactions (including social isolation, stabling 24/7)	4.55 (1.29)
5	Poor pasture management	4.50 (1.00)	Racing horses on the road (usually by gypsy/ travellers)	4.55 (1.44)	Lack of biosecurity and disease surveillance	4.30 (1.51)
6	Lack of understanding of horse welfare needs by owner/carer	4.5 (1.24)	Delayed euthanasia decisions	4.46 (1.04)	Delayed euthanasia decisions	4.28 (1.43)
7	Over-rugging (horse lacks control if too hot/cold)	4.42 (0.99)	Lack of biosecurity and disease surveillance	4.46 (1.13)	Lack of routine health care (e.g., vaccinations, dental, parasite prevention)	4.25 (1.14)
8	Lack of biosecurity and disease surveillance	4.42 (1.24)	Long-distance transport	4.42 (1.38)	Large worm burdens	4.23 (0.93)
9	Poor handling methods	4.33 (1.30)	Inability to perform normal social interactions (including social isolation, stabling 24/7)	4.37 (1.08)	Poor owner knowledge of horse care	4.21 (1.21)
10	Poorly fitting and restrictive tack	4.25 (1.06)	Overworking	4.36 (1.36)	Poor pasture management	4.18 (1.36)
11	Poor quality information available (online or from others)	4.25 (1.36)	Poorly fitting and restrictive tack	4.33 (1.44)	Over-rugging (horse lacks control if too hot/cold)	4.18 (1.47)
12	Unstable social groups (herd constantly changing)	4.08 (1.38)	Abandonment or stray horses	4.27 (1.19)	Unsuitable diets for equine feeding behaviour (e.g., feeding methods of stabled horses, turned out 24/7)	4.18 (1.29)
13	Inability to perform normal social interactions (e.g., social isolation, stabling 24/7)	4.00 (1.23)	Unsuitable diets for equine feeding behaviour (e.g., feeding methods of stabled horses, turned out 24/7)	4.26 (1.23)	Inappropriate use of food supplements	4.10 (1.10)
14	No regulation of establishments or service providers	3.97 (1.47)	Overweight riders for horse	4.25 (1.14)	Poor hoof care	4.10 (1.10)
15	Unsuitable diets for equine feeding behaviour (e.g., feeding methods of stabled horses, turned out 24/7)	3.93 (1.51)	Use of inappropriate training and handling	4.22 (1.44)	Neglect of sub-optimal care	4.09 (1.04)
16	Negative affective states (fear, frustration, boredom, depression, anxiety)	3.92 (1.38)	Unavoidable aggressive social interactions (e.g., restricted resources in groups)	4.13 (1.44)	No regulation of establishments or service providers	4.09 (1.48)
17	Poor pasture management	3.90 (1.00)	Neglect or sub-optimal care	4.08 (1.31)	Negative affective states (fear, frustration, boredom, depression, anxiety)	4.09 (0.99)
18	Neglect or sub-optimal care	3.78 (1.01)	Tethering without provision of shade, water etc.	4.08 (1.68)	Lack of understanding of horse welfare needs by owner/carer	4.07 (1.79)
19	Horse passports not fit for purpose	3.77 (1.35)	Large worm burdens	4.00 (1.29)	Overweight riders for horse	4.00 (1.18)
20	Overworking	3.73 (1.29)	Poor hoof care	4.00 (1.10)	Poorly fitting and restrictive tack	4.00 (1.34)
21	Poor health knowledge by owner	3.69 (0.80)	Hiring horses to unsuitable riders	3.95 (1.21)	Use of inappropriate training and handling	3.93 (1.35)
22	Poor weaning methods	3.64 (1.50)	Poor weaning methods	3.92 (1.56)	Poor weaning methods	3.91 (1.92)
23	Delay in seeking veterinary care	3.62 (1.33)	Poor health knowledge by owner	3.92 (1.64)	Indiscriminate breeding (including of compromised animals)	3.91 (1.48)
24	Overweight riders for horse	3.58 (1.31)	No regulation of establishments or service providers	3.91 (1.31)	Unstable social groups (herd constantly changing)	3.82 (1.33)
25	Inappropriate use of food supplements	3.58 (1.31)	Over-rugging (horse lacks control if too hot/cold)	3.83 (1.34)	Tethering without access to shade, water etc.	3.73 (1.62)
26	Indiscriminate breeding (including of compromised animals)	3.46 (1.31)	Poor owner knowledge of horse care	3.82 (1.24)	Abandonment or stray horses	3.70 (1.42)
27	Poor hoof care	3.45 (0.93)	Horse passports not fit for purpose	3.82 (1.40)	Poor health knowledge by owner	3.70 (1.79)
28	Being ridden	3.42 (1.93)	Negative affective states (fear, frustration, boredom, depression, anxiety)	3.77 (1.62)	Lack of access to clean/fresh water	3.64 (1.69)
29	Fear, stress or injury from use in work, sport or entertainment	3.35 (1.53)	Delay in seeking veterinary care	3.75 (1.60)	Hiring horses to unsuitable riders	3.54 (1.03)
30	Lack of routine health care (e.g., vaccinations, dental, parasite prevention)	3.23 (0.93)	Unstable social groups (herd constantly changing)	3.67 (1.50)	Unavoidable aggressive social interactions (e.g., restricted resources in groups)	3.46 (1.51)
31	Hiring horses to unsuitable riders	3.17 (1.47)	Poor pasture management	3.65 (1.17)	Long-distance transport	3.45 (1.37)
32	Poor transport of horses on domestic journeys	3.14 (1.12)	Lack of understanding of horse welfare needs by owner/carer	3.64 (1.37)	Fear, stress or injury from use in work, sport or entertainment	3.44 (1.18)
33	Long-distance transport	3.00 (1.41)	Fear, stress or injury from use in work sport or entertainment	3.63 (1.00)	Flygrazing (grazing horses without landowners permission)	3.40 (1.78)
34	Unavoidable aggressive social interactions (e.g., restricted resources in groups)	3.00 (1.04)	Indiscriminate breeding (including of compromised animals)	3.62 (1.06)	Overworking	3.34 (1.19)
35			Lack of routine health care (e.g., vaccinations, dental, parasite prevention)	3.42 (0.90)	Lack of easy access to health care	3.09 (0.98)
36			Poor transport of horses on domestic journeys	3.31 (1.37)		
37			Lack of easy access to health care	3.25 (1.31)		

**Table 3 animals-10-00647-t003:** Ranked prioritized welfare issues for horses derived from the workshop for individual horses (combined severity and duration of the welfare issue in the experts’ opinion) and for the UK horse population (perceived prevalence).

Rank Order	Horse Population (Perceived Prevalence)	Individual Horses (Severity + Duration)
1	Lack of biosecurity and disease surveillance	Delayed euthanasia decisions
2	Delayed euthanasia decisions	Lack of recognition of pain behaviour
3	Lack of understanding of horse welfare needs by owner/carer	Large worm burdens
4	Fear/stress/frustration from use in work, sport or entertainment	Obesity
5	Obesity	Unsuitable diets for equine feeding behaviour
6	Indiscriminate/inappropriate breeding	Hunger
7	Poorly fitting and restrictive tack	Inability to perform normal social interactions
8	Unstable social groups	Negative affective states
9	Unsuitable diets for equine feeding behaviour	Overworking
10	Poor weaning methods	Overweight riders

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
