# Peer review of "Determining a Welfare Prioritization for Horses Using a Delphi Method"

_animals, 2020, doi:10.3390/ani10040647_

Round 1
Reviewer 1 Report
Dear authors,
I found this work wonderful, with very interesting data, and a very intersting discussion. You introduction and discussion are long, but the information that you provide in each part had me attached to the paper from the very beginning.
I have a few minor comments that I believe to be important to a better comprehension of the paper.
Keywords:
Line 37 - horse has the h in bold, please correct it.
Results:
Line 168:
Table 2 - I believe that in the title you should add a recall of the score you have applied (Likert scale 1-6).
Still on table 2 - I have found some boxes with very similar welfare issues, which I believe to be the same. If they are the same, please modify them, so the text would be the same for the same welfare issue. I will list them to you using the rank order:
rank 2 - use of inappropriate training methods - see rank 15 and 21
rank 6 - Lack of understanding of horse welfare needs by owner/carer - see rank 18 and 32
rank 12 - unstable social groups (herd constantly changing) - see rank 24 and 30
rank 13 - inappropriate use of food supplements - see rank 25
rank 21 and 23 - poor health knowledge by owner - see rank 27
Line 180 to 183 - I did not understand if this agreement should unanimous or if there ias percentage of agreement. If there is a percentage of agreement, please mention it. If an unanimous agreement should be reached, please define it in the material and methods section.
Discussion:
Line 368 - please correct the brakets
Conclusion:
This is only a suggestion. I belive the paragraph you have in lines 389-396 should be in the discussion section. I think that you should not conclude your work with this paragraph to the reader. When I was reading it seems to me that you are minimizing your incredible work by placing this paragraph in the end of the paper. In my opinion you should move this paragraph to the general discussion (lines 190-240). This way you would have discussed the limitations of your study, but you would have a conclusion that represents your work, which is in my opinion well structured, with important information and important conclusions.
Author Response
Keywords:
Line 37 - horse has the h in bold, please correct it.
Response: Corrected
Results:
Line 168:
Table 2 - I believe that in the title you should add a recall of the score you have applied (Likert scale 1-6).
Responses: Added to Table legend
Still on table 2 - I have found some boxes with very similar welfare issues, which I believe to be the same. If they are the same, please modify them, so the text would be the same for the same welfare issue. I will list them to you using the rank order:
rank 2 - use of inappropriate training methods - see rank 15 and 21
rank 6 - Lack of understanding of horse welfare needs by owner/carer - see rank 18 and 32
rank 12 - unstable social groups (herd constantly changing) - see rank 24 and 30
rank 13 - inappropriate use of food supplements - see rank 25
rank 21 and 23 - poor health knowledge by owner - see rank 27
Response: Thank you – these have been corrected.
Line 180 to 183 - I did not understand if this agreement should unanimous or if there ias percentage of agreement. If there is a percentage of agreement, please mention it. If an unanimous agreement should be reached, please define it in the material and methods section.
Response: Added: Consensus at the workshop was considered to be unanimous where a final list was produced with the agreement of all participants. At Line 141-142.
Discussion:
Line 368 - please correct the brakets
Response: apologies but we cannot see where the correction is required?
Conclusion:
This is only a suggestion. I belive the paragraph you have in lines 389-396 should be in the discussion section. I think that you should not conclude your work with this paragraph to the reader. When I was reading it seems to me that you are minimizing your incredible work by placing this paragraph in the end of the paper. In my opinion you should move this paragraph to the general discussion (lines 190-240). This way you would have discussed the limitations of your study, but you would have a conclusion that represents your work, which is in my opinion well structured, with important information and important conclusions.
Response: Thank you – we were not sure it truly fitted elsewhere so have instead chosen to strength the final sentence, which is our belief that this approach and comparison with other studies has been sufficient to overcome the issues.
Reviewer 2 Report
This is a very important paper in the field of equine welfare linking practical expert knowledge to theoretical ideas of equine welfare. Delphi method is also a very promising method that creates interesting results. I have a major suggestion for changes for the tables and minor changes for the rest of the paper.
Major
Tables 1, 2 and 3 - There are discrepancies between the wording of the "welfare issues" in the three tables. Some of the Welfare issues in Table 2 and 3 are named differently in Table 1. As an example, row 1 in Table 2, obesity has a somewhat different clarifying text in Table 1 than in Table 2, lack of recognition of pain behavior in Table 2 is merged from two welfare issues in Table 1.
There are also discrepancies between Table 3 and Table 2. For example, row 4 in Table 3 "Fear/stress/frustration..." has different wording than row 29, left column in Table 2.
I suggest that the authors go through all welfare issues and check the wording in order for more clarity in the method.
Is it helpful to have an index number or letter for each welfare issue?
What happens with "Category of concern" (Table 1) in the text and Table 2 and 3. Could that be used for further clarity so that it becomes clear where the "welfare issues" belongs for example in the tables and lines 192-197.
Supplementary material - Why is there fewer rows in Table 1 in the Supplementary material than in Table 2 in the paper, shouldn't there be the same?
Minor
Line 58: omit Hartmann et al.
Line 90: consider adding a reference to a description of the project.
Line 140: Change "finial" to "final"
Line 181: "was more succesful" How was this measured?
Discussion: In the discussion I would like to see a section discussing how the difference in anonymity of the participants between the first (Anonymous) and second stage (face-to-face) might influence the results.
Line 371-374: here the authors could add reference to studies indicating that single-stabled horses develop sterstereotypies.
Author Response
Major
Tables 1, 2 and 3 - There are discrepancies between the wording of the "welfare issues" in the three tables. Some of the Welfare issues in Table 2 and 3 are named differently in Table 1. As an example, row 1 in Table 2, obesity has a somewhat different clarifying text in Table 1 than in Table 2, lack of recognition of pain behavior in Table 2 is merged from two welfare issues in Table 1.
There are also discrepancies between Table 3 and Table 2. For example, row 4 in Table 3 "Fear/stress/frustration..." has different wording than row 29, left column in Table 2.
I suggest that the authors go through all welfare issues and check the wording in order for more clarity in the method.
Response: Thank you here and in response to Reviewer 1 we have corrected and clarified between Tables 1 and within Table 2. For Table 3 the experts in the Workshop did amend the wording/combined some areas thus Table 3 is not an exact replica of Tables 1 and 2, this explanation has been added to the Methods.
Is it helpful to have an index number or letter for each welfare issue?
Response: Initially we did use this method, but it became difficult when experts felt that some categories were too similar and could be put together, so this has been removed from the MS.
What happens with "Category of concern" (Table 1) in the text and Table 2 and 3. Could that be used for further clarity so that it becomes clear where the "welfare issues" belongs for example in the tables and lines 192-197.
Response: This was part of the emergent coding process and arose from the categorisation during the Thematic analysis. It was part of the derivation of the Welfare issues to consider but was not presented to our experts in this way and therefore was not included in the Tables, as we did not wish to influence their decision making.
Supplementary material - Why is there fewer rows in Table 1 in the Supplementary material than in Table 2 in the paper, shouldn't there be the same?
Response: in order to reduce the workload on the experts (at their request, and to help retain experts in all rounds) we combined a small number of cases from round 1 where the experts had indicated that they believed the issues to be the same. This led to a small reduction in the number of issues presented to experts in the 2nd round compared to those in the 1st round. This has been indicated in the Table legend for the Supplementary Material.
Minor
Line 58: omit Hartmann et al.
Response: corrected in line with Request from Reviewer 3 also.
Line 90: consider adding a reference to a description of the project.
Response: The full project manuscript is in the final stages of preparation for submission and at present there is no reference for this. Reference to other papers derived from the project are given in the text where appropriate.
Line 140: Change "finial" to "final"
Response: Corrected
Line 181: "was more succesful" How was this measured?
Response: In response to the comments of other reviewers a sentence has been added to methods to clarify the level of success required for the Workshop component.
Discussion: In the discussion I would like to see a section discussing how the difference in anonymity of the participants between the first (Anonymous) and second stage (face-to-face) might influence the results.
Response: Added a section on this as lines 199-204.
Line 371-374: here the authors could add reference to studies indicating that single-stabled horses develop sterstereotypies.
Response: added reference to a review article to support this point.
Reviewer 3 Report
The manuscript "Determining a welfare prioritization for horses using a Delphi method" presents expert opinion on welfare issues that are most important and may need greatest attention in the UK. A topic that increasingly gains interest worldwide.
The study is well described, the objectives and results are adequately described and compared with previous results. Limitations are also stated.
minor revisions:
line 46: delete e.g. or add an example
line 57: delete Hartmann et al. and leave only [11]
line 58: I wouldnt add e.g. if its only there for the reference
line 317: after Thus, please add comma (same for line 322)
Author Response
minor revisions:
line 46: delete e.g. or add an example
Response: deleted e.g.
line 57: delete Hartmann et al. and leave only [11]
line 58: I wouldnt add e.g. if its only there for the reference
Response: reference citations corrected as requested
line 317: after Thus, please add comma (same for line 322)
Response: corrected as requested throughout.